# Optimization of Sparse Cross Array Synthesis via Perturbed Convex Optimization

**DOI:** 10.3390/s20174929

**Published:** 2020-08-31

**Authors:** Boxuan Gu, Yaowu Chen, Rongxin Jiang, Xuesong Liu

**Affiliations:** 1Institute of Advanced Digital Technology and Instrumentation, Zhejiang University, Hangzhou 310027, China; 11415011@zju.edu.cn; 2Engineering Research Center of Embedded Systems Education Department, Zhejiang University, Hangzhou 310027, China; 3Zhejiang Provincial Key Laboratory for Network Multimedia Technologies, Zhejiang University, Hangzhou 310027, China; rongxinj@zju.edu.cn; 4State Key Laboratory of Industrial Control Technology, Zhejiang University, Hangzhou 310027, China; 11015006@zju.edu.cn

**Keywords:** phased array 3-D imaging sonar system, sparse cross array, compressed sensing, perturbed convex optimization, multi-frequency algorithm

## Abstract

Three-dimensional (3-D) imaging sonar systems require large planar arrays, which incur hardware costs. In contrast, a cross array consisting of two perpendicular linear arrays can also support 3-D imaging while dramatically reducing the number of sensors. Moreover, the use of an aperiodic sparse array can further reduce the number of sensors efficiently. In this paper, an optimized method for sparse cross array synthesis is proposed. First, the beamforming of a cross array based on a multi-frequency algorithm is simplified for both near-field and far-field. Next, a perturbed convex optimization algorithm is proposed for sparse cross array synthesis. The method based on convex optimization utilizes a first-order Taylor expansion to create position perturbations that can optimize the beam pattern and minimize the number of active sensors. Finally, a cross array with 100 + 100 sensors is employed from which a sparse cross array with 45 + 45 sensors is obtained via the proposed method. The experimental results show that the proposed method is more effective than existing methods for obtaining optimum results for sparse cross array synthesis in both the near-field and far-field.

## 1. Introduction

Real-time 3-D sonar imaging technology is one of the most important innovations in underwater applications in recent years [1,2,3]. The phased array 3-D imaging sonar system transmits acoustical pulse signals penetrating the entire underwater detection scene, receiving sonar echo signals through a large planar array. The phased array technology simultaneously generates entire beam intensity signals to obtain real-time 3-D images [4]. With the development of underwater technology, an increased number of array sensors is required for better image quality. However, the high cost, power consumption, and computational complexity brought about by a large number of array sensors impede the practical implementation of this technology [5].

To reduce the number of array sensors, redundant sensors can be eliminated. In [6], several array configurations were proposed for analysis and comparison. These array configurations effectively reduce the number of array sensors. Among these configurations, a cross array with two perpendicular linear arrays has been employed in some sonar systems [7,8]. Experiments show that this cross array configuration can produce a beam pattern (BP) similar to that of a planar array [9]. However, the cross array sonar system requires considerable time to scan the entire detection range, resulting in a low frame rate. That is, the real-time performance of the system is poor. In [9], a multi-frequency (MF) algorithm is applied to the cross array configuration. The MF algorithm subdivides the vertical direction into several sectors. In each sector, a series of multiple frequency acoustical signals is transmitted sequentially, thereby significantly reducing the number of transmissions and increasing the frame rate. Using this method, a planar array with *M* × *N* sensors can be reduced to a cross array with only *M* + *N* sensors.

In addition, the use of sparse array techniques [10] can be applied to the cross array to further reduce the number of array sensors. A sparse array is an aperiodic array with a reduced number of sensors. By means of repositioning the remaining sensors and changing the weights, the desired BP can be maintained. Sparse array designs are grouped into two general categories [11]: (1) stochastic optimization and (2) deterministic optimization. In stochastic optimization, the sidelobe peak (SLP), main lobe width (MLW), beam pattern shape, etc., are chosen as the objective functions [12], after which a stochastic optimization algorithm is utilized to produce a solution that is optimal according to certain criteria. Using iterative techniques, the objective function gradually converges to the optimal solution. Commonly used stochastic optimization algorithms are simulated annealing algorithms [12,13,14], genetic algorithms [15], ant colony optimization [16], and particle swarm optimization [17]. However, due to the randomness inherent to these techniques, the stochastic optimization algorithms yield different results for each run. Furthermore, these algorithms are sensitive to variation in the initial values. Consequently, stochastic optimization algorithms have difficulty achieving global optimization. Multiple experiments as well as parameter adjustments are necessary to obtain optimal results [18].

The deterministic optimization algorithm approaches sparse array synthesis as a minimum *l*_0_-norm problem. Since the minimum *l*_0_-norm is non-convex and difficult to solve, the deterministic optimization algorithm approximates it by a similar, but easier to solve, problem [19]. Compared with stochastic optimization algorithms, these methods can solve the sparse array synthesis more effectively [18]. Recently, deterministic optimization algorithms have been developed on the basis of the compressed sensing (CS) theory. The main CS-based algorithms are Bayesian compressive sampling (BCS) algorithms [20,21,22], focal under-determined system solver (FOCUSS) algorithms [23,24], and convex optimization algorithms [25,26,27]. Moreover, some shaped beam patterns (such as flat-top BP and asymmetric sidelobe BP) are employed in sparse array synthesis via CS-based methods [28,29,30].

However, in some existing CS-based algorithms, the candidate sensor positions are constrained to initial discrete positions, which cannot ensure the degree of freedom for candidate sensor positions. To obtain better sparse array results, a denser initial sensor array is required to enhance the degree of freedom, which results in a large amount of calculation. In [23], first-order Taylor expansion is introduced to create position perturbations. Through this method, sensors can be placed in continuous positions instead of being placed in discrete grid positions.

In this paper, a perturbed convex optimization (PCO) method is proposed to synthesize a cross array in the near-field and far-field. The proposed method synthesizes the sparse array via iterative reweighted *l*_1_ minimization and uses a first-order Taylor expansion to create the position perturbations. The PCO method enhances the degree of freedom for candidate sensor positions, thus obtaining better sparse array results and BP performance. A sparse cross array in a 3-D imaging sonar system is designed via the PCO method. The sonar system is characterized by multiple frequency acoustical transmitting signals ranging from 205 to 300 kHz and a 50*λ*_min_ aperture at 300 kHz. An MF algorithm is applied to achieve near-field and far-field beamforming, with the near-field divided into several focus sections to simplify the calculation. The experimental results are presented and compared with those of other methods to verify the effectiveness of the proposed method.

This paper is organized as follows. In Section 2, a simplified beamforming of cross arrays for near-field and far-field is presented, and a PCO method is proposed and analyzed. In Section 3, a sparse cross array for a 3-D imaging sonar system is designed and simulated to evaluate the efficiency of the proposed method. In Section 4, experimental results are discussed. In Section 5, conclusions are drawn.

## 2. Methods

A cross array for a low-complexity real-time 3-D sonar imaging system consists of two perpendicular linear arrays [6], as shown in Figure 1. The receiving array, comprising *N* array sensors, is oriented in the horizontal direction, while the transmitting array, comprising *M* array sensors, is oriented in the vertical direction. The spacing of the array sensors of the transmitting array and the receiving array are *d_x_* and *d_y_*, respectively.

### 2.1. Multi-Frequency Cross Array Beamforming in the Near-Field and Far-Field

The cross array has the same effective 3-D acoustic imaging capability as the two-dimensional (2-D) planar array [7]. Under the same sonar signal frequency and array aperture condition, while a 2-D planar array requires *M* × *N* sensors, the cross array can obtain the same angular resolution with only *M* + *N* sensors, yielding a tremendous reduction in the number of array sensors. The main factor of the cross array that allows for such a large reduction in the sensor number is the orientation of the transmitting and receiving arrays with respect to each other, with the transmitting and receiving arrays performing beamforming in the vertical and horizontal directions, respectively. Through the joint action of the transmitting and receiving arrays in this configuration, the 3-D acoustic image is constructed.

In conventional cross array systems, the transmitting array sequentially transmits an acoustical signal to a predetermined sequence of *Q* vertical beam directions [8]. For each predetermined vertical beam direction, the receiving array receives the acoustical echo signals and performs beamforming in the *P* horizontal directions within the beam range. When all the vertical beam direction transmissions are completed and the horizontal receiving beamforming calculations have been performed, a complete 3-D acoustic image can be formed. The beam distribution diagram is shown in Figure 2.

The cross array sonar system requires considerable time to scan the entire detection range, which leads to a low frame rate and poor real-time performance of the system. In [9], a multi-frequency (MF) algorithm is proposed on the basis of a cross array to improve the real-time performance.

If it is assumed that the number of vertical transmitting beam directions is *Q* and the number of horizontal receiving beam directions is *P*, the specific process of the MF algorithm is described as follows. First, the set of vertical beam directions is divided into *K* different sectors. Within each sector, the sensor array is transmitted in J = *Q*/*K* different preset vertical beam directions. Through the phase shift compensation between the array sensors, the acoustical signals of different frequencies are sequentially transmitted to the preset *J* vertical beam directions, with each frequency (from *f*_1_ to *f_J_*) corresponding to a vertical beam direction. Subsequently, after the acoustical signal transmission of all frequencies in the sector completes, the receiving array receives the acoustical echo signal, and the beamforming calculation is performed in the frequency domain to generate *P* × *J* beam intensity results. The process is repeated for each sector yielding the complete *P* × *Q* beam intensity results. Figure 3 shows the transmitting process of the MF algorithm.

In the far-field where distance exceeds *D*^2^/*λ* [4], the transmitting and receiving beamforming can be regarded as the beamforming of two linear arrays, with the BP given respectively as follows:(1)$${\mathrm{BP}}_{T}\left(u,\;\lambda \right)\;=\;\left|\overset{M}{\underset{m=1}{\sum }}w_{m}\;\exp\left(i\frac{2\pi }{\lambda }{ux}_{m}\right)\;\right|$$
(2)$${\mathrm{BP}}_{R}\left(v,\;\lambda \right)\;=\;\left|\overset{N}{\underset{n=1}{\sum }}w_{n}\;\exp\left(i\frac{2\pi }{\lambda }{vy}_{n}\right)\;\right|$$
where *u* = sin*α*-sin*α*_0_, *v* = sin*β* − sin*β*_0_; *x_m_* = (*m* − (*M* + 1)/2)*d_x_* gives the sensor positions for the transmitting array; *y_n_* = (*n* − (*N* + 1)/2)*d_y_* gives the sensor positions for the receiving array; *w_m_* and *w_n_* are the weights of the transmitting and receiving sensors, respectively; *λ* is the acoustical wavelength; (*α*, *β*) is the arrival direction, and (*α*_0_, *β*_0_) is the steering direction.

The BP of the cross array in the far-field can be regarded as the product of the transmitting BP and the receiving BP [9], as follows:(3)$${\mathrm{BP}}_{C}\left(u,\;v,\;\lambda \right)\;=\;\left|\sum_{m=1}^{M}w_{m}\;\exp\left(i\frac{2\pi }{\lambda }{ux}_{m}\right)\;\right|\cdot \left|\sum_{n=1}^{N}w_{n}\;\exp\left(i\frac{2\pi }{\lambda }{vy}_{n}\right)\;\right|.$$

The conventional near-field beamforming algorithm differs at different distances, which leads to a higher computational burden. Furthermore, the optimization of sparse cross arrays in the near-field requires huge computational cost to fulfill the conditions required to cover the entire near-field. In Zhao et al. [31], to simplify the near-field BP calculation, distances in the near-field are divided into several focus regions, and the focal distance *r*_0_ of each focus region is selected as shown in Figure 4. Through this simplification, the optimized transmitting and receiving BP of a cross array in the near-field can be approximated as follows:(4)$${\mathrm{BP}}_{\mathrm{OT}}\left(u,\;\lambda ,\;\delta \right)\;=\;\left|\overset{M}{\underset{m=1}{\sum }}w_{m}\;\exp\left(i\frac{2\pi }{\lambda }{(ux}_{m}\;+\;\delta \frac{x_{m}^{2}}{2})\right)\;\right|$$
(5)$${\mathrm{BP}}_{\mathrm{OR}}\left(v,\;\lambda ,\;\delta \right)\;=\;\left|\overset{N}{\underset{n=1}{\sum }}w_{n}\;\exp\left(i\frac{2\pi }{\lambda }{(vy}_{n}\;+\;\delta \frac{y_{n}^{2}}{2})\right)\;\right|$$
(6)$$\delta =\;\frac{1}{r}-\;\frac{1}{r_{0}}$$
where *r* is the distance between the object and the array center, and *r*_0_ is the distance between the focal and the array center.

The optimized BP of a cross array can be regarded as the product of the transmitting beamforming and the receiving beamforming as follows:(7)$${\mathrm{BP}}_{O}\left(u,\;v,\;\lambda ,\;\delta \right){\;=\;\mathrm{BP}}_{\mathrm{OT}}\left(u,\;\lambda ,\;\delta \right)\;\cdot {\;\mathrm{BP}}_{\mathrm{OR}}\left(v,\;\lambda ,\;\delta \right).$$

It is shown that when *r* = *r*_0_, the near-field beamforming is equal to far-field beamforming, and when the quantity |*δ*| is large, the near-field beam pattern distortion becomes problematic. To better satisfy the BP constraints, we impose the following maximum [31]:(8)$${\left|\delta \right|}_{\max}\;=\;\frac{{2\lambda }_{\min}}{D^{2}}.$$

The simplified beamforming stays the same in each focus region, which greatly reduces the computational complexity of the 3-D imaging sonar system. At the same time, the BP constraint on the entire near-field and far-field can be achieved by constraining the entire *δ* in the sparse cross arrays synthesis, which is easier to accomplish in the case of convex optimization.

### 2.2. Sparse Cross Array Synthesis Method

#### 2.2.1. Iterative Reweighted l_1_ Minimization

The synthesis of a sparse cross array can be regarded as an *l*_0_-norm problem as follows:(9)$${\mathrm{variable}\;w\;\min\;∥w∥}_{0}\;s.t{.\;\mathrm{BP}}_{O}\;\mathrm{under}\;\left(\mathrm{BP}.C\right)$$
where *w* = [*w_m_ w_n_*]; *w_m_* and *w_n_* are the weight matrices of the transmitting and receiving arrays; ${∥w∥}_{0}\;$ is the *l*_0_-norm of the ***w*** matrix, i.e., the number of non-zero elements of *w*. BP.C represents the BP constraints including SLP, MLW (at −3 dB), and the beam pattern shape shown in Figure 5 [18]. The solution *w* is a sparse matrix: non-zero elements are active, and zero elements are inactive.

This optimization problem is very difficult to solve directly because the minimum *l*_0_-norm is non-convex. According to the CS theory, the optimization problem of Equation (9) can be approximated by the following iterative reweighted *l*_1_ minimization problem based on a convex optimization algorithm [32]:(10)$$\mathrm{variable}w^{i}\;\min\;∥w^{i}\;◦\rho^{i}∥{}_{1}\;s.t{.\;\mathrm{BP}}_{O}\;\mathrm{under}\;\left(\mathrm{BP}.C.\right)$$
(11)$$\rho^{i}{\;=\;1/(w}^{i-1}\;+\;\epsilon )$$
where ${∥\;w∥}_{1}$ is the *l*_1_-norm of the ***w*** matrix, which is the sum of the absolute values of all elements in ***w***; *w^i^* ◦ *ρ^i^* is the Hadamard product of the two matrices *w^i^* and *ρ^i^*; *i* is the number of iterations; and *ρ* is the coefficient related to the optimization result of the last iteration, which makes the minimum *l*_1_-norm problem of Equation (10) gradually approximate the minimum *l*_0_-norm problem of Equation (9). Moreover, in the minimum *l*_1_-norm, the value of *w* cannot be equal to zero, but it approaches zero, and the elements less than 1 × 10^−6^ in magnitude can be considered as zero elements [18]; ϵ is slightly less than the minimum value of *w*, which ensures that the zero elements are likely to be non-zero in the next iteration. In the first iteration, ρ^1^ is set to a matrix of all ones; a MATLAB software for disciplined convex programming CVX [33] is used to solve the minimum *l*_1_-norm problem of Equation (10) to obtain *w*^1^ and determine ϵ, which is slightly less than the minimum value of *w*^1^. In the following iteration, ρ*^i^* is obtained using Equation (11), and the minimum *l*_1_-norm problem of Equation (10) is then solved to obtain *w^i^* until the sparse array results converge.

CVX is a modeling framework for solving disciplined convex problems, including linear and quadratic programs, semidefinite programs, *l*_1_-norms, etc. CVX is implemented in Matlab, conveniently solving constrained norm minimization, entropy maximization, and many other convex optimization problems. The general convex optimization problems can be expressed in the following form:(12)$${\min\;f}_{0}\left(x\right)\;s.t{.\;f}_{i}\left(x\right)\leq 0,\;i=1,\ldots ,M$$
where *x* is the objective variable, *f*_0_ is the objective function, and *f*_1_, …, *f_M_* are the constraint functions.
(13)$$f_{0}\left(x\right)=Cx\;A_{i}{x=b}_{i},\;i=1,\ldots ,M$$
where *C*, *A_k_* and *b_i_* are given matrices. The dual problem associated with Equation (12) is solved as follows:(14)$${\max\;b}^{T}y\;\sum_{i=1}^{M}y_{i}A_{i}\;+\;z\;=\;C$$
where *y_i_* and *z* are variables.

SDPT3 [34] is the default solver of CVX to solve convex optimization problems. SDPT3 is a primal-dual interior-point algorithm via the path-following paradigm. In each iteration of the algorithm, a predictor search direction is calculated to decrease the duality gap as much as possible. The solver uses two search directions: the Helmberg–Kojima–Monteiro (HKM) direction [35,36,37] and the Nesterov–Todd (NT) direction [38]. Then, the algorithm generates a Mehrotra-type corrector step [39] to approach the central path. The algorithm does not impose any neighborhood restrictions and tries to achieve feasibility and optimality simultaneously.

*x*^0^, *y*^0^ and *z*^0^ are initialized in the first iteration. Suppose the variables in the current and the next iterations are (*x*, *y*, *z*) and (*x*^+^, *y*^+^, *z*^+^) respectively. The step-length parameter in the current and the next iterations are (*α*, *β*, *γ*) and (*α*^+^, *β*^+^, *γ*^+^). Set *γ*^0^ = 0.9. The iteration stops if the relative duality gap (relgap) is less than 1 × 10^−8^.
(15)$$\mathrm{relgap}\;=\;\frac{xz}{1+\max\left(\left|Cx\right|,\left|b^{T}y\right|\right)}$$

(*x*^+^, *y*^+^, *z*^+^) are set as following:(16)$$x^{+}\;=\;x\;+\;\alpha \Delta {x,\;y}^{+}\;=\;y\;+\;\beta \Delta {y,\;z}^{+}\;=\;z\;+\;\beta \Delta z$$
(17)$$A^{T}\Delta y\;+\;\Delta z\;=\;c-z-A^{T}y\;A\Delta x\;=\;b-Ax$$
(18)$$\alpha \;=\;\min\left(1,\;\frac{-\gamma }{E_{\min}\left(x^{-1}\Delta x\right)}\right),\;\beta \;=\;\min\left(1,\;\frac{-\gamma }{E_{\min}\left(z^{-1}\Delta z\right)}\right)$$
where (Δ*x*, Δ*y*, Δ*z*) are search directions. *E*_min_(*x*^-1^Δ*x*) is the minimum eigenvalue of (*x*^-1^Δ*x*). Set *γ*^+^ = 0.9 + 0.09 min(*α*, *β*).

The search directions (Δ*x*, Δ*y*, Δ*z*) are obtained via the symmetrized Newton equation with respect to an invertible matrix *P*. If semidefinite blocks are present, the HKM direction is selected; otherwise, the NT direction is selected. The HKM direction is corresponding to *P* = *z*^1/2^; the NT direction is corresponding to *P* = *N*^−1^, where *N^T^zN* = *N*^−1^
*× N*^−*T*^.

Problems that can be solved by CVX must be disciplined convex problems, and CVX is not efficient for very large problems (for example, a very large sparse planar array synthesis). For the problem of this paper, CVX is an effective solution.

#### 2.2.2. Perturbed Convex Optimization

To enhance the degree of freedom for candidate sensor positions, a PCO method is proposed to optimize sparse array synthesis. The beamforming can be approximated as in Equations (19) and (20) using first-order Taylor expansion [23].
(19)$${\mathrm{BP}}_{\mathrm{OT}}\left(x\;+\;\Delta x\right)\;\approx {\;\mathrm{BP}}_{\mathrm{OT}}\left(x\right)\;+\;\Delta x\;{\mathrm{dBP}}_{\mathrm{OT}}(x)dx$$
(20)$${\mathrm{BP}}_{\mathrm{OR}}\left(y+\;\Delta y\right)\;\approx {\;\mathrm{BP}}_{\mathrm{OR}}\left(y\right)\;+\;\Delta y\;{\mathrm{dBP}}_{\mathrm{OR}}(y)dy$$
where ${\mathrm{dBP}}_{\mathrm{OT}}(x)dx$ is the derivative of BP_OT_ with respect to *x*; |Δx| < *d*_min_/2 and |Δy| < *d*_min_/2 are the position perturbations; and *d*_min_ is the minimum distance between sensors. On the basis of the first-order Taylor expansion, a PCO method is proposed to optimize the position perturbation and weight simultaneously. For the transmitting array, the optimization solves the following PCO problem to find the optimal position perturbation and weight.
(21)$${\mathrm{variable}\;v}^{i}{\;\min\|w}^{i}\;◦{\;\rho }^{i}{\|}_{1}\;s.t{.\;\mathrm{BP}}_{\mathrm{OT}}\left(x\;+\;\Delta x^{i}\right)\;\mathrm{under}\;\left(\mathrm{BP}.C.\right)\;{\&\&\;w}^{i}\;\cdot {\;d}_{\min}/2\;-\;\left|s^{i}\right|\;>\;0$$
where *v^i^* = [*w^i^ s^i^*]; *s^i^* = *w^i^* ∙ △*x^i^*; and *i* is the number of iterations. The matrix *v^i^* contains the position perturbation and weight information. The PCO method obtains the minimum number of active array sensors, while the BP satisfies the constraint, and the position perturbations are constrained within *d*_min_/2. Moreover, in the sonar system, the sensor positions are fixed, but the weight can vary under different conditions. To obtain better BP performance under different conditions (transmitting frequency and *δ*), Equation (22) can be solved independently at different transmitting frequencies and *δ*.
(22)$$\mathrm{variable}\;w\;\min\;\mathrm{SLP}\;s.t{.\;\mathrm{BP}}_{O}\;\mathrm{under}\;\left(\mathrm{BP}.C\right)$$

Through this method, sensors can be placed in continuous positions instead of being placed on discrete grid points. The proposed method provides more degree of freedom for the sensors.

As shown in Equation (7), the beamforming of the cross array can be regarded as the product of the transmitting beamforming and the receiving beamforming. In addition, the transmitting beamforming and receiving beamforming can be regarded as the beamforming of two linear arrays. Therefore, sparse cross array synthesis can be divided into two sparse linear array syntheses. The flow diagram of a sparse cross array synthesis via the PCO method is shown in Figure 6. The procedure is described as follows:

The transmitting array of *M* sensors is considered. The transmitting frequency is set from *f*_1_ to *f_J_*. *δ* is set from *δ*_1_ to *δ_A_* (−*δ*_min_ to +*δ*_max_). In the first iteration, $\rho_{m}^{1}$ is set to a matrix of all ones, and CVX [33] is employed to solve the PCO problem of Equation (21) to obtain △*x*^1^ and $w_{m}^{1}$, which makes the BP_OT_ satisfy the constraints over the entire frequency range, as well as the near-field and far-field conditions. *ϵ* is determined to be slightly less than the minimum value of $w_{m}^{1}$. In the following iterations, the PCO problem is solved to obtain △*x^i^* and $w_{m}^{i}$ and iterated until the number of active sensors remains unchanged for five iterations. At this point, the iterations are concluded, and the positions and weight values of the sparse transmitting array are considered optimal. Next, Equation (15) is applied to optimize the BP performance under different conditions.

The receiving array of *N* sensors is synthesized in the same way as the transmitting array. Since the PCO method is deterministic optimization, the results of the sparse receiving array are the same as those of the transmitting array when *M* = *N* at the same conditions.

## 3. Results

### 3.1. Sparse Cross Array Synthesis

A cross array with 100 + 100 sensors is employed, and a sparse cross array is synthesized using the proposed methods. We compare the sparse array result with those in [14] and [31]. The transmitting frequency ranges from 205 to 300 kHz, with frequency steps of 5 kHz. The sensor spacing is *λ*_min_/2 = 2.5 mm at 300 kHz. The critical distance between the near-field and far-field is *D*^2^/*λ*_min_ = 12.5m. The near-field is divided into three sections: 2–3 m, 3–5.5 m, and 5.5–12.5 m. The focal points are at 2.4 m, 4 m, and 9 m, respectively. Therefore, *δ* is within the range of –0.083 to 0.083, which satisfies Equation (6) (2*λ*_min_/*D*^2^ = 0.16). The (u, v) space is set within (−1 to 1, −1 to 1) which is divided equally into 400 × 400 beams. The SLP is set to –23 dB. The main lobe in (*u*, *v*) is restricted to within 0.022 at 300 kHz. The optional positions are expanded into 500 + 500 with sensor spacing of *λ*_min_/10. The parameter *ϵ* is set to 0.004.

Through the proposed method, a sparse cross array with 45 + 45 sensors was achieved. The BP satisfies the constraints well in both the near-field and the far-field. The experimental codes and results are attached in Appendix A. Table 1 provides a comparison of the sparse array results between the proposed method and those proposed in [14,31].

Compared with the existing sparse cross array synthesis methods, the proposed method obtains a smaller number of active sensors and achieves better BP performance. Moreover, the sparse cross array syntheses in [14,31] are based on the simulated annealing (SA) algorithm and thus produce a different result each time they are executed. In these methods, the number of iterations required to obtain an optimal solution is large and unpredictable. The method proposed in this paper is not stochastic in nature and thus does not suffer from these shortcomings.

The BPs of the sparse cross array under different conditions are shown in Figure 7. In Figure 7a, *δ* is 0 and the transmitting frequency is 300 kHz; in Figure 7b, *δ* is 0.083 and the transmitting frequency is 300 kHz; in Figure 7c, *δ* is 0 and the transmitting frequency is 205 kHz; and in Figure 7d, *δ* is 0.083 and the transmitting frequency is 205 kHz.

The transmitting BPs under different conditions are shown in Figure 8. In Figure 8a, *δ* is 0 and the transmitting frequency is 300 kHz; the SLP is −23.67 dB and the MLW is 1.22°. In Figure 8b, *δ* is 0.083 and the transmitting frequency is 300 kHz; the SLP is −23.67 dB and the MLW is 1.22°. In Figure 8c, *δ* is 0 and the transmitting frequency is 205 kHz; the SLP is −23.82 dB and the MLW is 1.75°. In Figure 8d, *δ* is 0.083 and the transmitting frequency is 205 kHz; the SLP is −23.82 dB and the MLW is 1.75°. The experiment shows that in the same cross array, the SLP and MLW are related to the transmitting frequency, but not to *δ*.

The positions and weight values of the optimized active sensors at 300 kHz in the far-field are shown in Figure 9 and Table A1 in Appendix B; the minimum spacing between sensors is 0.889*λ*_min_ at 300 kHz; the array aperture of the proposed method is 249.42 mm (49.88*λ* at 300 kHz), which is slightly larger than those of the methods in [14,31]. Therefore, the proposed method obtains higher angular resolution. The number of active sensors in the transmitting array versus the number of iterations is shown in Figure 10.

### 3.2. Flat-Top BP Synthesis

A linear array with flat-top BP is employed, and a sparse linear array is synthesized using the proposed methods. We compare the sparse array result with those in [29]. The aperture and sensor spacing of the initial linear array are 14λ and 0.7λ, respectively. The main beam width is 40° and the SLP of the shaped beam is constrained less than −35 dB. The parameter ϵ is set to 0.01.

Through the proposed method, a sparse linear array with 18 sensors was achieved. The beam width (at −3 dB) is 41.4°, and the SLP is −35.27 dB. Table 2 provides a comparison of the sparse array results between the proposed method and those proposed in [29].

The flat-top BP of the sparse linear array is shown in Figure 11. The positions and weight values of the optimized active sensors are shown in Table 3.

### 3.3. Asymmetric Sidelobe BP Synthesis

A linear array with asymmetric sidelobe BP is employed, and a sparse linear array is synthesized using the proposed methods. We compare the sparse array result with those in [29]. The aperture and sensor spacing of the initial linear array are 12λ and 0.6λ, respectively. The left SLP of the shaped beam is constrained less than −35 dB and the right SLP of the shaped beam is constrained less than −25 dB. The parameter *ϵ* is set to 0.01.

Through the proposed method, a sparse linear array with 14 sensors was achieved. The beam width (at −3 dB) is 6.21°. The left SLP is −37.01 dB and the right SLP is −26 dB. Table 4 provides a comparison of the sparse array results between the proposed method and those proposed in [29].

The asymmetric sidelobe BP of the sparse linear array is shown in Figure 12. The positions and weight values of the optimized active sensors are shown in Table 5.

## 4. Discussion

The experimental results demonstrate that the proposed method is efficient for synthesizing a sparse cross array in the near-field and far-field compared with the existing methods. The proposed method introduces position perturbations via first-order Taylor expansion and optimizes the sensor position and weight simultaneously. The proposed method enhances the degree of freedom for candidate sensor positions; thus, the sparse array results and BP performance achieved are better than those of existing methods. In Figure 9, since the PCO method is a deterministic optimization and *M* = *N* = 500, the results of sparse transmitting and receiving arrays are equal under the same condition. In Figure 7 and Figure 8, the experimental results demonstrate that the BPs stay the same in the near-field and far-field at the same transmitting frequency. BPs are related to transmitting frequency, but not to *δ*. In Equation (15), *w_m_* is optimized independently at different *λ* and *δ* values. Referring to Equation (4), when *w_m_* satisfies Equation (23) at the same *λ*, BP_OT_ is the same at different *δ* values (near-field and far-field).
(23)$$w_{m}\left(x_{m},\;\delta ,\;\lambda \right){\;=\;w}_{m}{(x}_{m},\;0,\;\lambda )\cdot \;\exp(-i\frac{{\pi x}_{m}^{2}}{\lambda }\delta )$$

Table 1. provides the positions and weight values of the sparse transmitting array at 300 kHz (*λ* = 5mm) in the far-field (*δ* = 0). Based on Equations (4) and (23), in Figure 8a,b, the transmitting BPs are the same in the near-field and far-field at 300 kHz. Therefore, the experimental results verify the effectiveness of the proposed method.

## 5. Conclusions

In this paper, an optimized method of sparse cross arrays synthesis was proposed and used to design a 3-D sonar system. An MF algorithm was utilized to accomplish near-field and far-field beamforming, with the near-field divided into several focus regions to simplify the calculation. A PCO method was proposed for the synthesis of the aperiodic sparse cross array. The optimization method is based on an iterative reweighted *l*_1_ minimization algorithm and uses first-order Taylor expansion to create the position perturbations of the sparse array to enhance the degree of freedom for candidate sensor positions. The experimental results show that the proposed method obtains optimum results for sparse cross array synthesis in both the near-field and far-field.

## Figures and Tables

**Figure 1 sensors-20-04929-f001:**
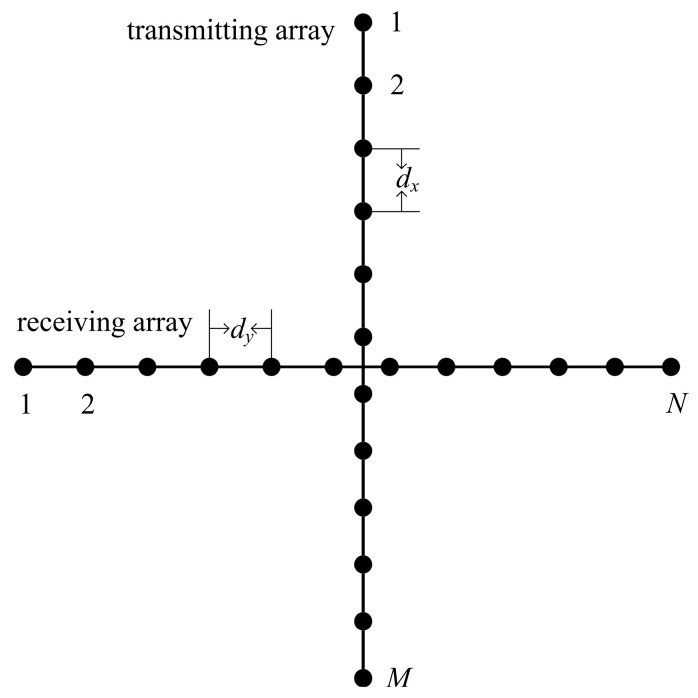
Configuration of the cross array.

**Figure 2 sensors-20-04929-f002:**
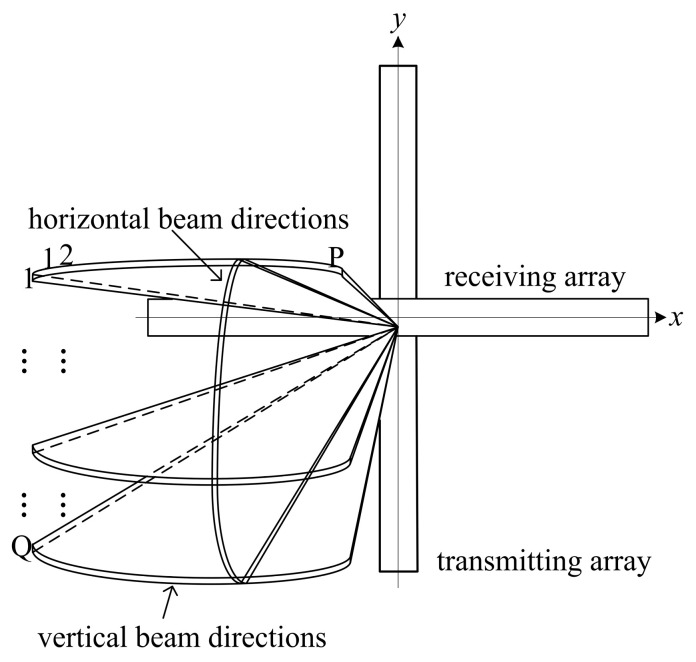
Beam distribution diagram.

**Figure 3 sensors-20-04929-f003:**
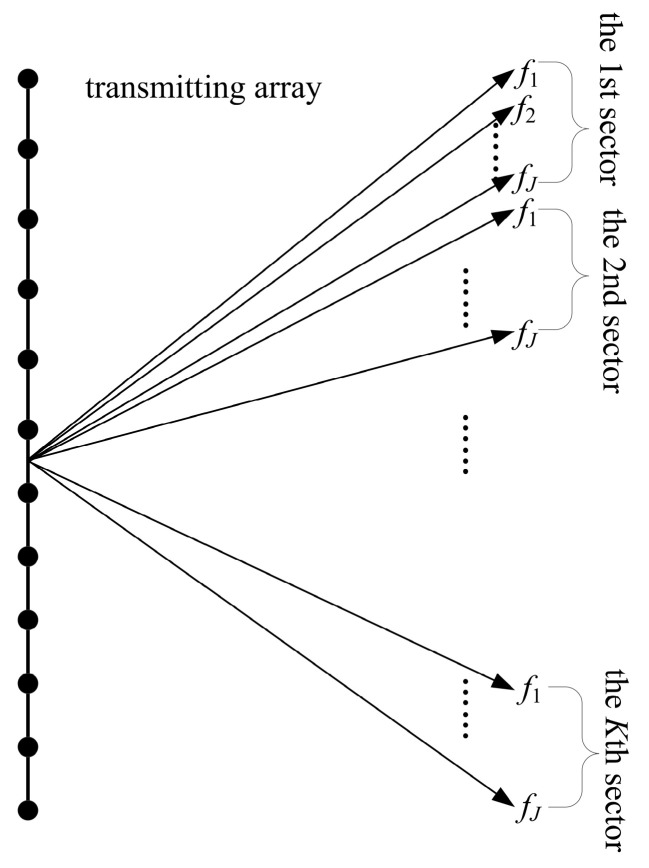
Transmitting process of the multi-frequency (MF) algorithm.

**Figure 4 sensors-20-04929-f004:**
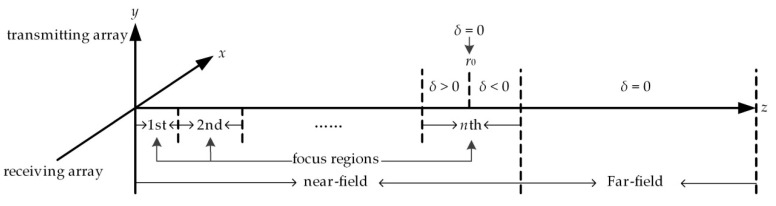
Division of the near-field.

**Figure 5 sensors-20-04929-f005:**
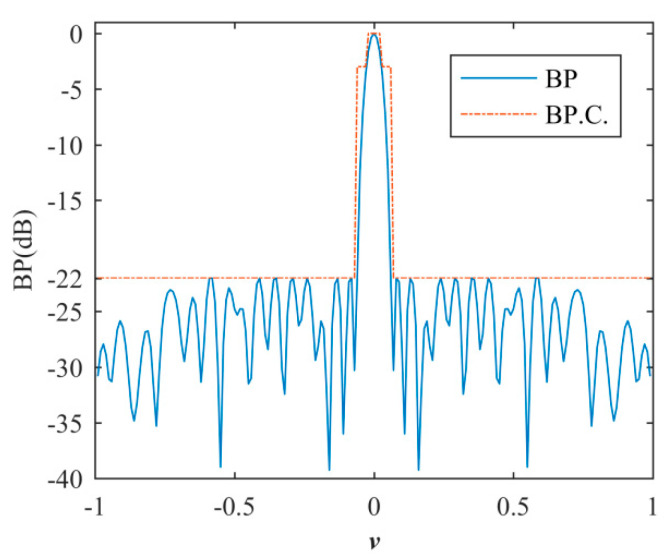
Beam pattern (BP) constraints.

**Figure 6 sensors-20-04929-f006:**
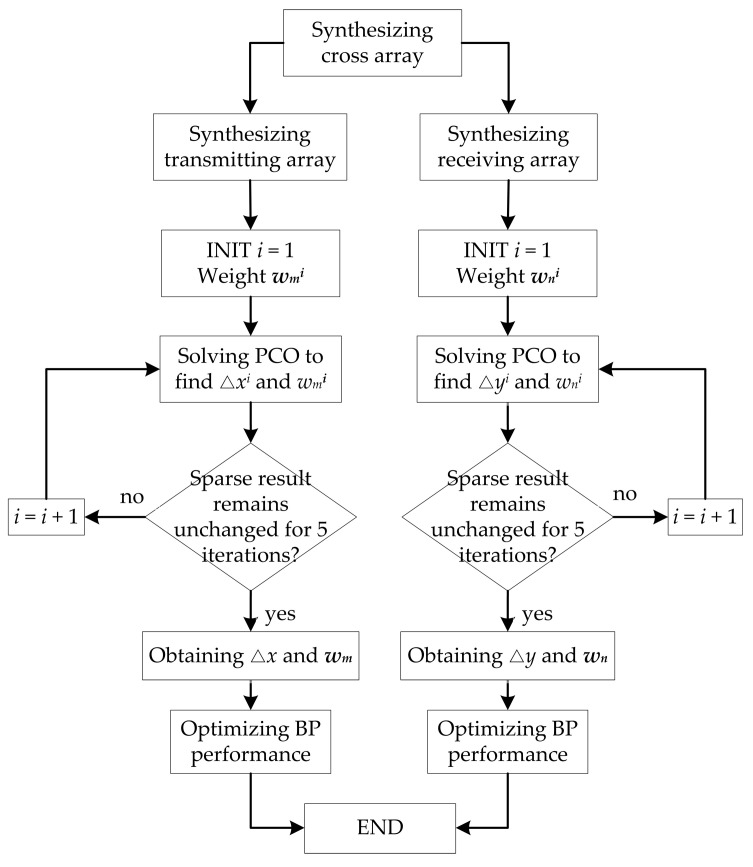
Flow diagram of a sparse cross array synthesis in near-field and far-field.

**Figure 7 sensors-20-04929-f007:**
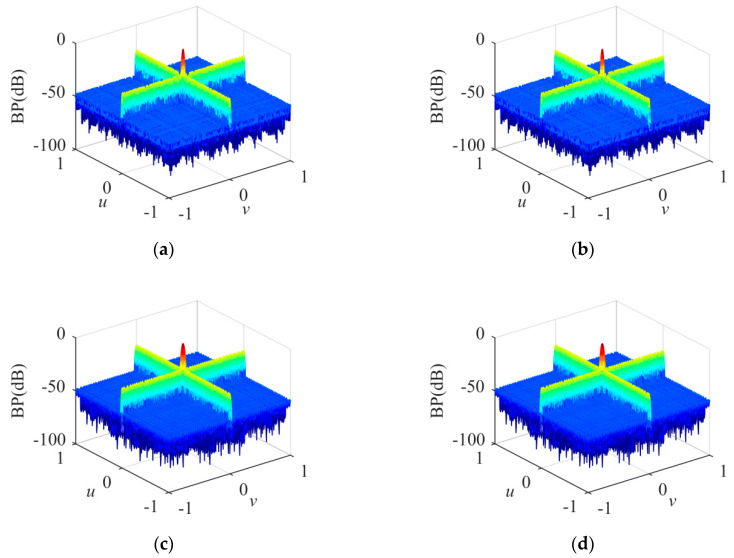
BPs of sparse cross array. (**a**) *δ* = 0, transmitting frequency = 300 kHz; (**b**) *δ* = 0.083, transmitting frequency = 300 kHz; (**c**) *δ* = 0, transmitting frequency = 205 kHz;(**d**) *δ* = 0.083, transmitting frequency = 205 kHz.

**Figure 8 sensors-20-04929-f008:**
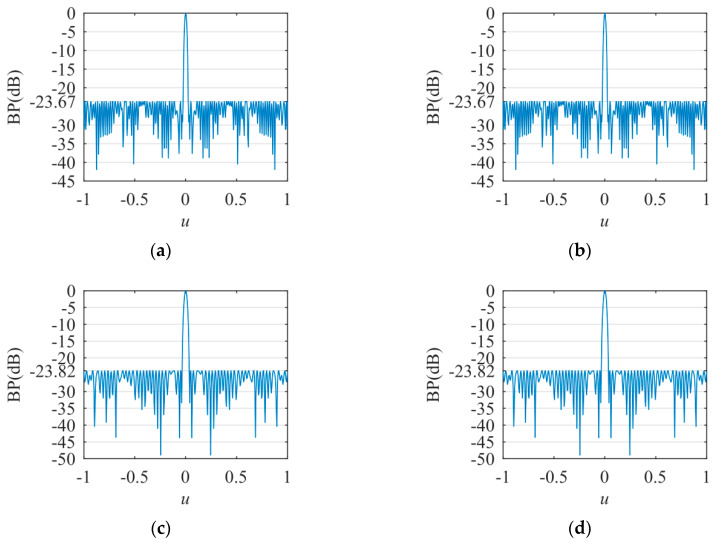
The transmitting BPs. (**a**) *δ* = 0, transmitting frequency = 300 kHz; (**b**) *δ* = 0.083, transmitting frequency = 300 kHz; (**c**) *δ* = 0, transmitting frequency = 205 kHz; (**d**) *δ* = 0.083, transmitting frequency = 205 kHz.

**Figure 9 sensors-20-04929-f009:**
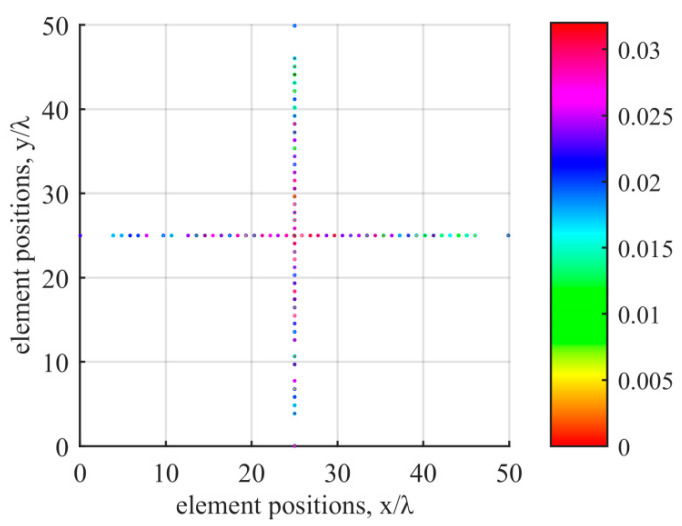
Positions and weight values of active sensors.

**Figure 10 sensors-20-04929-f010:**
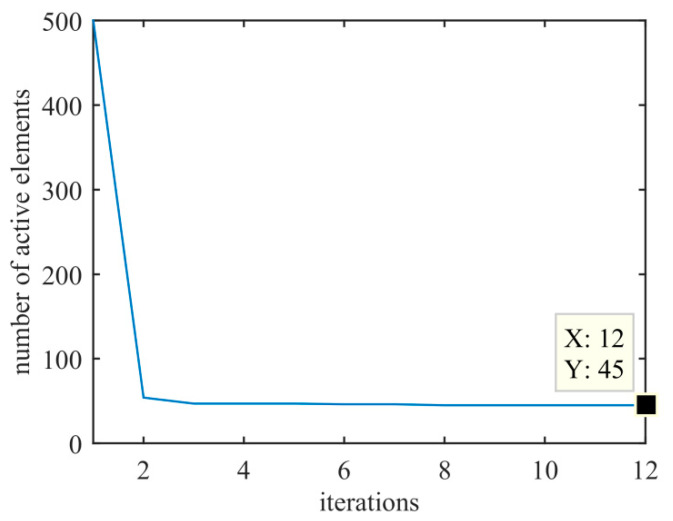
Number of active sensors versus the number of iterations.

**Figure 11 sensors-20-04929-f011:**
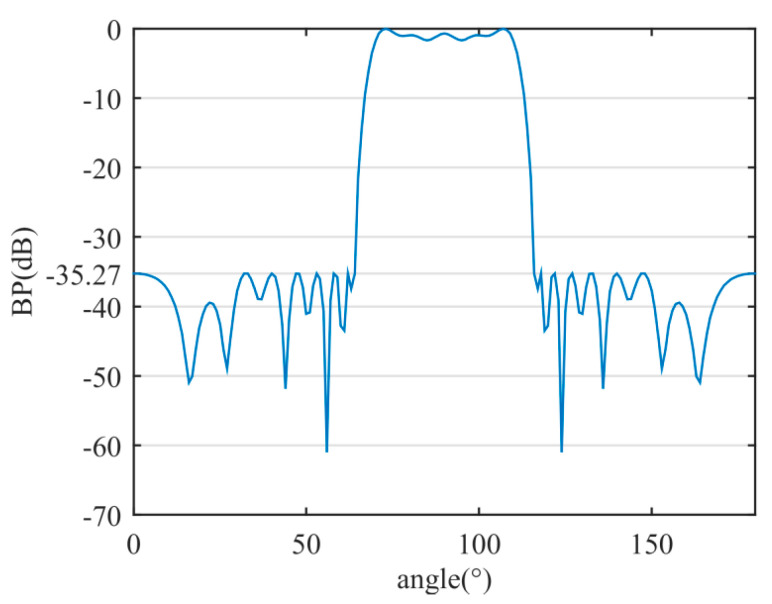
The flat-top BP of the sparse linear array.

**Figure 12 sensors-20-04929-f012:**
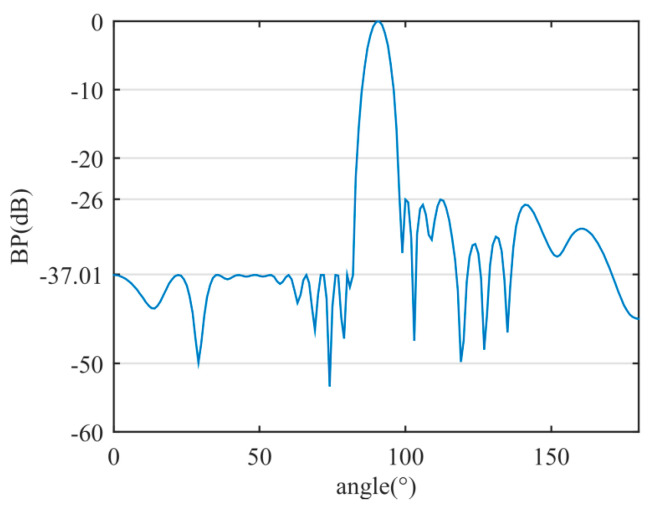
The asymmetric sidelobe BP of the sparse linear array.

**Table 1 sensors-20-04929-t001:** Comparisons of sparse cross array synthesis results among the proposed method and the existing methods.

Method		*Ns* ^1^	SLP (dB)		RES (°) ^2^
			Near-Field	Far-Field	
Liu et al. [14]		110	−18.7	−22	1.28
Zhao et al. [31]	1st optimization	118	−21.61	−21.9	1.28
	2nd optimization	107	-	−22	1.28
Proposed method		90	−23.67	−23.67	1.22

*Ns*^1^: number of active sensors; RES ^2^: angular resolution at 300 kHz. SLP: sidelobe peak.

**Table 2 sensors-20-04929-t002:** Comparisons of flat-top BP synthesis results among the proposed method and the existing methods.

Method	*Ns* ^1^	SLP (dB)	Beam Width (°)
Liang et al. [29]	20	−33.25	41.0
proposed method	18	−35.27	41.4

*Ns*^1^: number of active sensors.

**Table 3 sensors-20-04929-t003:** Positions and weight values of the sparse linear array with flat-top BP.

Position (λ)	Weight	Position (λ)	Weight	Position (λ)	Weight
0.48	0.0080	4.93	−0.0685	9.05	−0.0753
0.78	0.0011	5.25	−0.0421	10.00	0.0169
1.14	0.0266	6.34	0.2849	10.40	0.0479
2.31	−0.0266	6.99	0.4608	11.68	−0.0272
3.58	0.0454	7.64	0.2938	12.85	0.0145
3.98	0.0225	8.74	−0.0365	13.26	0.0115

**Table 4 sensors-20-04929-t004:** Comparisons of asymmetric sidelobe BP synthesis results among the proposed method and the existing methods.

Method	*Ns* ^1^	Left SLP (dB)	Right SLP (dB)	Beam Width (°)
Liang et al. [29]	20	−37.02	−25.46	6.88
proposed method	14	−37.01	−26.00	6.21

*Ns*^1^: number of active sensors.

**Table 5 sensors-20-04929-t005:** Positions and weight values of the sparse linear array with asymmetric sidelobe BP.

Position (λ)	│Weight│	Weight (rad)	Position (λ)	│Weight│	Weight (rad)
0.00	0.005	0.1284	5.57	0.1292	−0.0293
0.81	0.0188	0.5582	6.42	0.1225	−0.0334
1.51	0.0387	0.3489	7.26	0.1094	−0.0626
2.28	0.0625	0.2164	8.08	0.0885	−0.1500
3.07	0.0842	0.0674	8.91	0.0641	−0.1971
3.92	0.1040	0.1162	9.72	0.0388	−0.2684
4.75	0.1232	0.0733	10.48	0.0264	−0.5216

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
