# Peer review of "Optimization of Sparse Cross Array Synthesis via Perturbed Convex Optimization"

_sensors, 2020, doi:10.3390/s20174929_

Round 1
Reviewer 1 Report
This paper proposes an optimization method of cross array synthesis. The novelty and procedure of the proposed method are presented. The effectiveness of the proposed method is also presented via the experimental results.
The paper is easy to follow because of good structure and polite explanations.
However, the following problems should be solved to more clarify the novelty and effectiveness of the proposed method.
[Major comments]
- The reference [29] is the authors' previous work and the method of [29] is an important role in this paper because this is compared with the proposed method. Despite that, the novelty compared with this previous paper is not presented in Introduction (only simple explanations on [29] is presented in Sec. 2), and it seems difficult for many readers to understand the differences between the proposed and conventional methods. Please clearly state the novelty compared with the previous work of [29] in Introduction.
- The comparison with the conventional methods is performed from the viewpoint of only the sidelobe peak. However, it seems that not only the peak level but also the shape of the sidelobe pattern is important in practical use. Please compare the sidelobe pattern of the proposed and some conventional methods to clarify the practicality of the proposed method.
[Minor comments]
- The parameter epsilon might be sensitive to the results of the proposed method. Please indicate how to set this parameter appropriately. How did you set epsilon=0.004 without prior information on the weights?
- In table 1, the angular resolution is also slightly improved using the proposed method compared with the conventional methods. Please discuss the reason for this.
Reviewer 2 Report
1)The authors claimed that “in CS-based algorithms, the candidate sensor positions are constrained to initial discrete positions, which cannot ensure the degree of freedom for candidate sensor positions”. Here, the CS-based algorithms refer to only some existing methods or all methods with CS-based?
2)There are some existing works, such as,
[1] Giulia Buttazzoni, et al. Synthesis of sparse arrays radiating shaped beams. 2016 IEEE International Symposium on Antennas and Propagation & USNC/URSI National Radio Science Meeting.
[2] Lei Liang, Can Jin, Hailin Li, Jianjiang Zhou. A hybrid algorithm of orthogonal perturbation method and convex optimization for beamforming of sparse antenna array. Apr 2020, ELECTROMAGNETICS.
[3] Mei-yan Zheng, et al. Sparse Planar Array Synthesis Using Matrix Enhancement and Matrix Pencil. International Journal of Antennas and Propagation, 2013.
These works should be discussed and the main contributions of the manuscript should be stated clearly.
3) In section 2.2, for the Sparse Cross Array Synthesis method, the minimum l0-norm is non-convex and converted to l1-norm problem. How to solve the l1-norm minimum is referred to reference [31]. But the reviewer thinks it should be discussed in detail here, to make the contents self-contained and easy-understanding. It is seemed that the optimization is the main theme of the manuscript. Moreover, any limitations of this solution?
4) The figure 9 is not clear and hard to figure out the theme.
5) More comparative experiments should be provided to validate the proposed method.
Round 2
Reviewer 1 Report
None
Author Response
We would like to take this opportunity to thank you for reviewing our manuscript.
Reviewer 2 Report
Most of the issues are cleared. But the response for the issue No.3 should be further improved.
the response only present a general description of cvx, not the detailed solution about the problem with CVX. For example,
1) the detailed expression of formula (12).
2) How to determine the search directions?
3) for the constraint: disciplined convex problem, is the problem expressed in formula (12) strictly matched? If not, what the situation is?
etc.
